# Stochastic logistic models reproduce experimental time series of microbial communities

**Lana Descheemaeker[1,2], Sophie de Buyl[1,2]\***

[1]Applied Physics Research Group, Physics Department, Vrije Universiteit Brussel, Brussel, Belgium; [2]Interuniversity Institute of Bioinformatics in Brussels, Vrije Universiteit Brussel Université Libre de Bruxelles, Brussels, Belgium

**Abstract** We analyze properties of experimental microbial time series, from plankton and the human microbiome, and investigate whether stochastic generalized Lotka-Volterra models could reproduce those properties. We show that this is the case when the noise term is large and a linear function of the species abundance, while the strength of the self-interactions varies over multiple orders of magnitude. We stress the fact that all the observed stochastic properties can be obtained from a logistic model, that is, without interactions, even the niche character of the experimental time series. Linear noise is associated with growth rate stochasticity, which is related to changes in the environment. This suggests that fluctuations in the sparsely sampled experimental time series may be caused by extrinsic sources.

**\*For correspondence:**
sdebuyl@vub.be

**Competing interests:** The authors declare that no competing interests exist.

## Introduction

Microbial communities are found everywhere on earth, from oceans and soils to gastrointestinal tracts of animals, and play a key role in shaping ecological systems. Because of their importance for our health, human-associated microbial communities have recently received a lot of attention. According to the latest estimates, for each human cell in our body, we count one microbe (*Sender et al., 2016*). Dysbiosis in the gut microbiome is associated with many diseases from obesity, chronic inflammatory diseases, some types of cancer to autism spectrum disorder (*Gilbert et al., 2016*). It is therefore crucial to recognize what a healthy composition is, and if unbalanced, be able to shift the composition to a healthy state. This asks for an understanding of the ecological processes shaping the community and dynamical modeling.

The dynamics of complex ecosystems can be studied by considering the number of individuals of each species, referred to as abundances, at subsequent time points. There are several ways to characterize experimental time series properties. Models typically focus on one specific aspect such as the stability of the community (*May, 1972*; *Coyte et al., 2015*; *Levine et al., 2017*; *Grilli et al., 2017*; *Gavina et al., 2018*; *Gibbs et al., 2018*), the neutrality (*Fisher and Mehta, 2014*; *Washburne et al., 2016*), or mechanisms leading to long-tailed rank abundance distributions (*Solé et al., 2002*; *Brown et al., 2002*; *McGill et al., 2007*; *Matthews and Whittaker, 2015*). Different types of dynamical models have been proposed. A first distinction can be made between neutral and non-neutral models. Neutral models assume that species are ecologically equivalent and that all variation between species is caused by randomness. In such models, no competitive or other interactions are included. A second distinction is made between population-level and individual-based models. Generalized Lotka-Volterra (gLV) models describe the system at the population level and assume that the interactions between species dictate the community's time evolution. Both deterministic and stochastic implementations exist for gLV models. Stochastic models include a noise term. There are multiple origins of the noise: intrinsic noise captures the fluctuations due to small

numbers, extrinsic noise models external factors such as changing immigration rates of species or changing growth rates mediated by a varying flux of nutrients. Individual-based or agent-based models include self-organized instability models (*Solé et al., 2002*) and the controversial neutral model of *Hubbell, 2001*; *Rosindell et al., 2011*. A classification scheme that assesses the relative importance of different ecological processes from time series has been proposed in *Faust et al., 2018*. The scheme is based on a test for temporal structure in the time series via an analysis of the noise color and neutrality. Applied to the time series of human stool microbiota, it tells us that stochastic gLV or self-organized instability models are more realistic. Here, we will however only focus on stochastic gLV models. The reason for this is twofold. First, one can encompass the whole spectrum of ecosystems from neutral to niche with gLV models (*Fisher and Mehta, 2014*). Second, we aim at describing dense ecosystems and even though an individual-based model might be more accurate, in the large number limit it will be captured by a Langevin approximation, that is, by the stochastic gLV model.

Our goal is to compare time series generated by stochastic gLV models with experimental time series of microbial communities. We aim at capturing all observed properties mentioned above—the rank abundance profile, the noise color, and the niche character—as well as the statistical properties of the differences between abundances at successive time points with one model. As is shown in Properties of experimental time series, the abundance distribution is heavy-tailed, which means that few species are highly abundant and many species have low abundances. Despite the large differences in abundances, the ratios of abundances at successive time points and the noise color are independent of these abundances and although the fluctuations are large, the results of the neutrality tests indicate that the experimental time series are in the niche regime. To sum up, we seek growth rates, interaction matrices, immigration rates, and an implementation of the noise in stochastic gLV models to obtain the experimental characteristics.

We simulated time series using gLV equations. The interaction matrices are random as was introduced by *May, 1972*. The growth rates are determined by the choice of the steady-state, which is set to either equal abundances for all species or abundances according to the rank abundance profiles found for experimental data. For the noise, we consider different implementations corresponding to different sources of intrinsic and extrinsic noise.

Our analysis constrains the type of stochastic gLV models able to grasp the properties of experimental time series. First, we show that there is a correlation between the noise color and the product of the mean abundance and the self-interaction of a species. The noise color profile for such models will, therefore, depend on the steady-state. This implies that imposing equal self-interaction strengths for all species, what can be done to ensure stability (*Fisher and Mehta, 2014*; *Gibson et al., 2016*), is incompatible with the properties of experimental time series. Second, from the differences between abundances at successive time points, we conclude that a model with mostly extrinsic (linear) noise agrees best with the experimental time series. Third, neutrality tests often result in the niche regime for time series generated by noninteracting species with noise. We, therefore, conclude that all stochastic properties of experimental time series are captured by a logistic model with large linear noise. However, interactions are not incompatible with those properties. This suggests using stochastic logistic models as null models to test for interactions. Our results go along the lines of the ones obtained by *Grilli, 2019* which state that the stochastic logistic model can be interpreted as an effective model capturing the statistics of individual species fluctuations.

All codes are available online (see Additional files: Code).

## Results

### Properties of experimental time series

We study time series from different microbial systems: the human gut microbiome (*David et al., 2014*), marine plankton (*Martin-Platero et al., 2018*), and diverse body sites (hand palm, tongue, fecal) (*Caporaso et al., 2011*; *Figure 1A*). A study of the different characteristics for a selection of these data is represented in *Figure 1*. The complete study of all time series can be found in *Supplementary file 1*: Analysis of experimental data. We propose a detailed description of the properties of the experimental time series. They fall essentially into two categories. The stability and rank abundance are tightly connected to the deterministic part of the equations while the

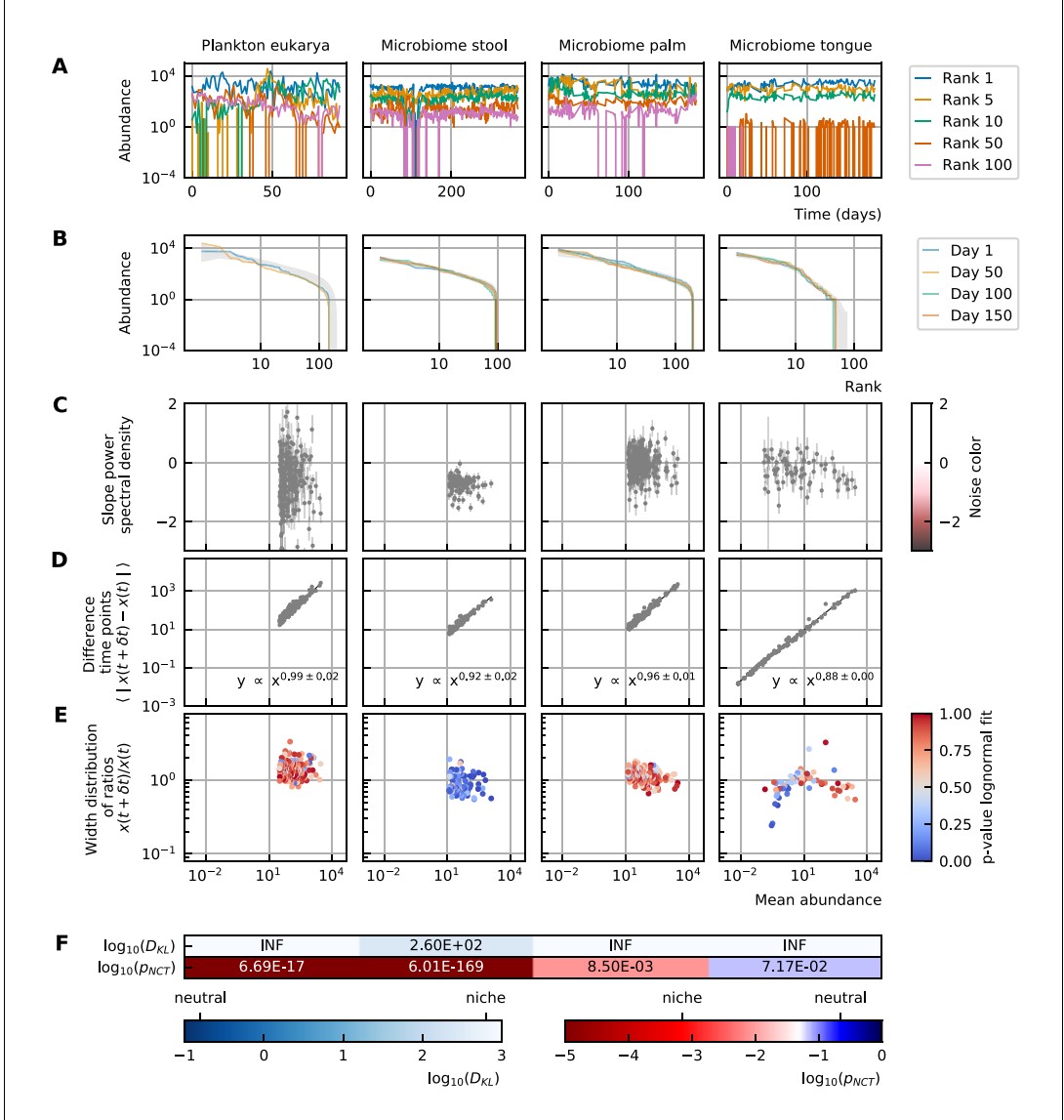

**Figure 1.** Characteristics of experimental data. (A) Time series. (B) Rank abundance profile. The abundance distribution is heavy-tailed and the rank abundance remains stable over time. (C) Noise color: No clear correlation between the slope of the power spectral density and the mean abundance of the species can be seen. The noise colors corresponding to the slope of the power spectral density are shown in the colorbar (white, pink, brown, black). (D) Absolute difference between abundances at successive time points: There is a linear correspondence (in log-log scale) between the mean absolute difference between abundances at successive time points and the mean abundance of the species. Because the slope is almost one, this hints at the linear nature of the noise. (E) Width of the distribution of the ratios of abundances at successive time points: The width of the distribution of successive time points is large (order 1) and does not depend on the mean abundance of the species. Most of the species fit well a lognormal distribution: the p-values of the Kolmogorov-Smirnov test are high. (F) Neutrality: The values of the Kullback-Leibler divergence ($D_{KL}$) and the neutral covariance test ($p_{NCT}$) are explicitly given. Additionally, we use color codes for both tests with the neutral regime represented by dark blue. White and red indicate the niche regime for the KL test and NCT respectively. We conclude that most experimental time series are in the niche regime.

differences between abundances at successive time points and noise color explain the stochastic behavior. The neutrality is more subtle and depends on the complete system.

## The time series show fluctuations over time

The experimental time series show large fluctuations over time. We can ask the question whether the origin of this variation is biological or technical, and assume that most of the variation can be

contributed to biological processes. This hypothesis is supported by the results of *Silverman et al., 2018* for microbial communities of an artificial gut. Here, the biological variation becomes five to six times more important than the technical variation for the sampling interval of a day. Also, *Grilli, 2019* shows the time correlation of experimental time series which is non-zero. In the case where the variation is mostly due technical errors, we expect to see no correlation. Because no experimental error-bars are available for most of the data and because we assume most variation has a biological origin, we did not consider the errors on the species abundances.

## The abundance distribution is heavy-tailed

The first aspect of community modeling that has been widely studied during the last years is the stability of the steady-states. Large random networks tend to be unstable (*May, 1972*). This problem is often solved by considering only weak interactions, sparse interaction matrices (*May, 2001*) or by introducing higher-order interactions (*Grilli et al., 2017*; *Gavina et al., 2018*; *Sidhom and Galla, 2019*). Although the stability of gLV models decreases with an increasing number of participating species, the stability only depends on the interaction matrix and not on the abundances (*Gibbs et al., 2018*). The abundance distribution of the experimental data is heavy-tailed. This means that there are few common and many rare species. The distribution of the steady-state values can also be represented by a rank abundance curve (see *Box 1B*). Although the abundances show large fluctuations over time, the rank abundance remains stable (*Figure 1B*).

## The differences between abundances at successive time points are large and linear with respect to the species abundance

Time series can be described by the differences between abundances at successive time points. We propose to focus on two specific representations of the information contained in those differences. First, we consider the mean absolute difference between abundances at successive time points $\langle \mid x(t + \delta t) - x(t) \mid \rangle$ as a function of the mean abundance $\langle x(t) \rangle$ (see *Box 1D*). For the experimental data, the relation between these variables is a monomial—this means that it is linear on the log-log scale (*Figure 1D*). The fact that the slope of this line is almost one hints at a linear nature of the noise.

Second, we examine the distribution of the ratios of the abundances at two successive time points $x(t + \delta t)/x(t)$ (see *Box 1E*). The width of this distribution tells how large the fluctuations are. To measure this width, we fit the distribution with a lognormal curve for which the mean is fixed to be one as the fluctuations occur around steady-state. For most of the species of experimental data (except for the stool data), the fit of the distribution to a lognormal curve is good (*Figure 1E*). Furthermore, we notice that the distribution is wide—in the order of 1—and that the width does not depend on the mean abundance of the species (*Figure 1E*).

## The noise color is independent of the mean abundance of the species

The noise of a time series can be studied by considering the distribution of the frequencies of the fluctuations. This distribution can be defined by its slope, which is interpreted as the noise color (see *Box 1C*). We notice that there is no correlation between the noise color and the mean abundance of the species for experimental time series (*Figure 1C*).

## Experimental time series are in the niche regime

In neutral theory, it is assumed that all species or individuals are functionally equivalent. It is challenging to test whether a given time series was generated by neutral or niche dynamics. We use two definitions of neutrality measures: the Kullback-Leibler divergence as used in *Fisher and Mehta, 2014* and the neutral covariance test as proposed by *Washburne et al., 2016* (see *Box 1F*). Both neutrality measures indicate that most experimental time series are in the niche regime (*Figure 1F*).

## Reproducing properties of experimental time series from stochastic generalized Lotka-Volterra models

We find that the aforementioned characteristics of experimental time series can be reproduced by stochastic logistic equations. We first explain how to choose the growth rate to obtain the heavy-tailed experimental abundance distribution. Next, we discuss how the noise color determines the

# Box 1. Definitions of the studied characteristics We study multiple characteristics of the dynamics of microbial communities.

We here define these characteristics. The labels (A-F) denote the different figures of *Figure 1* and *Figure 4*.

  A. A **time series** represents the time evolution of the abundances of different species of the community.

  B. The **rank abundance distribution** describes the commonness and rarity of all species. It can be represented by a rank abundance plot, in which the abundances of the species are given as a function of the rank of the species, where the rank is determined by sorting the species from high to low abundance. These curves can generally be fitted with power law, lognormal, or logarithmic series functions (*Limpert et al., 2001*; *McGill et al., 2007*; *Brown et al., 2002*).

  C. The **noise color** describes the distribution of the frequencies of the fluctuations of a time series of a species. It is defined by the slope of a linear fit through the power spectral density. White, pink, brown and black noise correspond to slopes around $0, -1, -2$ and $-3$ respectively. The more negative the slope is—this corresponds to darker noise—the more structure there is in the time series (*Faust et al., 2018*).

  D. We study the mean absolute **difference** between abundances at successive **time points** $\langle \, | \, x(t + \delta t) - x(t) \, | \, \rangle$ as a function of the mean abundance $\langle x(t) \rangle$. These values represent the jumps of the abundances from one time point to the next.

  E. We measure the **ratios of the abundances** at two successive time points $x(t + \delta t)/x(t)$. The advantage of this method is that it captures the direction of a jump between two time points: for ratios higher than one the jump is positive, for ratios lower than one the jump is negative. The distribution of these ratios fits a lognormal curve with a mean at one as the fluctuations occur around steady-state and the width of the distribution tells how large the fluctuations of a time series are. The goodness of the fit is defined by the p-value of the Kolmogorov-Smirnov test. Higher p-values denote a better fit. We use the width as a characteristic and compare the widths of different species. Examples of the fitted lognormal curve can be found in *Supplementary file 1*: Supporting results.

  F. The **Kullback-Leibler divergence** measures how different the multivariate distribution of species abundances is from a distribution constructed under the assumption of ecological neutrality. The idea of the **neutral covariance test** is to compare the time series with a Wright-Fisher process. A Wright-Fisher process is a continuous approximation of Hubbell's neutral model for a large and finite community. In particular, it tests the invariance with respect to grouping. More about the validity of these neutrality measures can be found in the *Supplementary file 1*: Supporting results.

self-interaction of a species given its abundance and how the implementation of the noise determines the slope of the mean absolute increment $\langle \, | \, x(t + \delta t) - x(t) \, | \, \rangle$ and the mean abundance $\langle x(t) \rangle$ (such as in *Figure 1D*). In the end, by using the appropriate choice for the self-interactions, growth rates, and noise implementation, we conclude that a stochastic logistic model can reproduce all the stochastic properties, including the niche regime for the neutrality tests although the model does not include any interactions.

## The rank abundance distribution can be imposed by fixing the growth rate

Random matrix models do typically not give rise to heavy-tailed abundance distributions. Neither is it known which properties of the interaction matrix and growth rates are required to obtain a realistic rank abundance distribution. We can however enforce the desired rank abundance artificially by solving the steady-state of the gLV equations. Given the steady-state abundance vector $\vec{x}^*$ and

interaction matrix $\omega$, we impose the growth rate $\hat{g} = -\omega \vec{x}^*$. One model that results in heavy-tailed distributions is the self-organized instability model proposed by *Solé et al., 2002*.

For logistic models, the growth rate is equal to the product of the self-interaction and mean abundance. The noise color and the width of the distribution of ratios $x(t + \delta t)/x(t)$ depend on this product. To obtain given characteristics—a predefined noise color and width of the distribution of ratios $x(t + \delta t)/x(t)$—the choice of the growth rate will dictate the choice of the remaining free parameters, the sampling time step $\delta t$ and the noise strength $\sigma$.

## The noise color is determined by the mean abundance and the self-interaction of the species

To study the noise color, we first consider a model where the species are not interacting. The noise color is independent of the implementation of the noise but depends on the product of the mean abundance and the self-interaction of the species (*Figure 2A*). For noninteracting species, the growth rate equals the product of the self-interaction and the steady-state abundance. Because we consider fluctuations around steady-state, the mean and the steady-state abundance are nearly equal and the x-axis of *Figure 2A*; *Figure 2B*; *Figure 2C*; can be interpreted as the growth rate. Also, the strength of the noise does not change its color (*Figure 2C*). A parameter that is important for the noise color is the sampling rate: the higher the sampling frequency the darker the noise

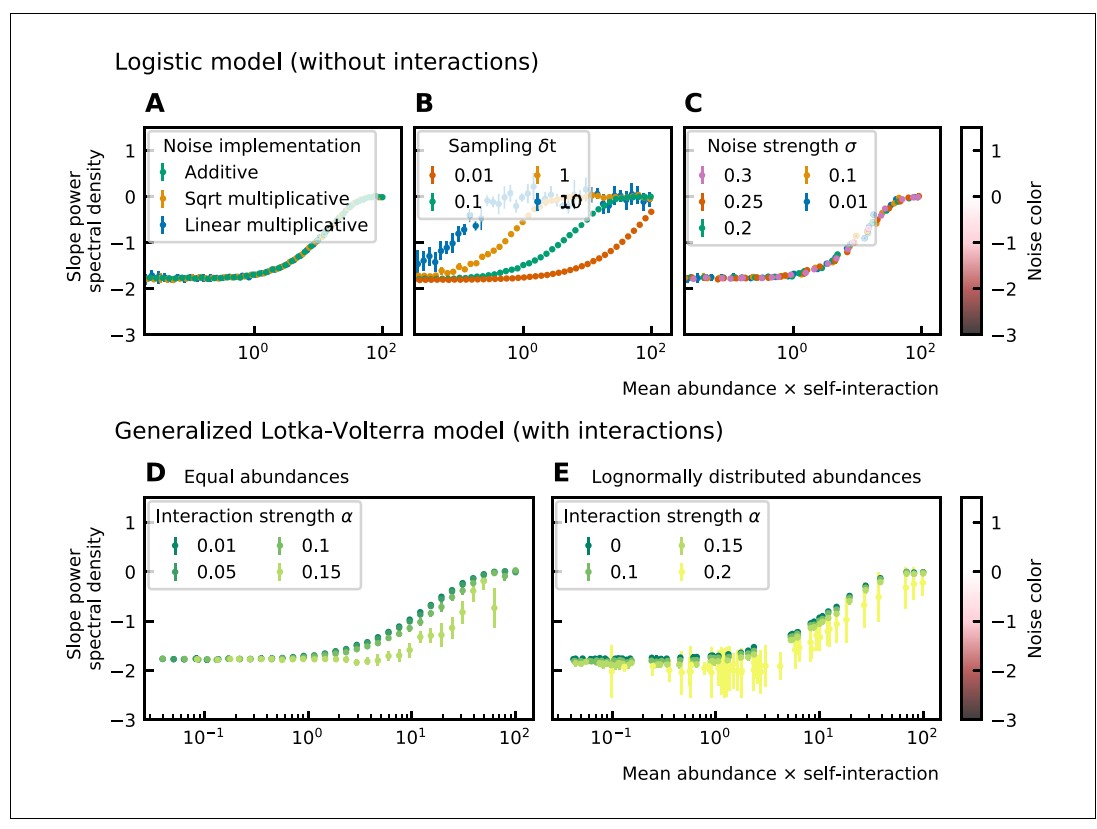

**Figure 2.** Noise color as a function of the mean abundance and self-interaction for stochastic logistic and gLV equations. The noise colors corresponding to the slope of the power spectral density are shown in the colorbar (white, pink, brown, black). The mean abundance determines the noise color when there is no interaction, the implementation method (**A**) and the strength of the noise (**C**) have no influence. A smaller sampling time interval $\delta t$, which is equivalent to a higher sampling rate, makes the noise darker (**B**). For gLV models with interactions, larger interaction strengths make the noise colors darker for systems with equal abundances (**D**) as well as systems with heavy-tailed abundance distributions (**E**).

becomes (*Figure 2B*). This is in agreement with the results of *Faust et al., 2018*. Darker noise corresponds to more structure in the time series. The more frequent the abundances are sampled the more details are visible and the underlying interactions become more visible. We conclude that the noise color is only dependent on the mean abundance, the self-interactions, and the sampling rate. Figures of the dependence on the mean abundance and self-interaction separately can be found in *Supplementary file 1*: Supporting results.

For interacting species, increasing the strength of the interactions makes the color of the noise darker in the high mean abundance range (*Figure 2D*; *Figure 2E*). Importantly, for interacting species with a lognormal rank abundance, the correlation between the noise color and mean abundance is preserved (*Figure 2E*). The data can be fit to obtain a bijective function between the product of the mean abundance and the self-interaction, and the noise color. Assuming this model is correct, we can obtain an estimate for the self-interaction coefficients given the mean abundance and noise color by fixing the sampling rate and the interaction strength. The uncertainty on the estimates is larger where the fitted curve is more flat (slopes of the power spectral density around $-1.7$ and 0), but many experimental values of the stool microbiome data lie in the pink region where the self-interaction can be estimated for this model.

## The implementation of the noise determines the correlation between the mean absolute increment $\langle \, | \, x(t + \delta t) - x(t) \, | \, \rangle$ and the mean abundance $\langle x(t) \rangle$

Next, we study the differences between abundances at successive time points (see *Figure 1D*). From the results of the noise color, we can estimate the self-interaction for the dynamics of the experimental data. We use the rank abundance and the self-interaction inferred from noise color of the microbiome data of the human stool to perform simulations and calculate the characteristics of the distribution of differences between abundances at successive time points. We here assume that there are no interactions. More results for dynamics with interactions are in *Supplementary file 1*: Supporting results. We first study the correlation between the mean absolute difference between abundances at successive time points $\langle \, | \, x(t + \delta t) - x(t) \, | \, \rangle$ and the mean abundance $\langle x(t) \rangle$. For linear multiplicative noise, the slope of the curve of the logarithm of the mean absolute difference between abundances at successive time points $\log_{10}(\langle \, | \, x(t + \delta t) - x(t) \, | \, \rangle)$ as a function of the logarithm of the mean abundance $\log_{10}(\langle x(t) \rangle)$ is one. For multiplicative noise that scales with the square root of the abundance, the slope is around 0.66 and for additive noise, the slope is zero. By combining both linear noise and noise that scales with the square root of the abundance, slopes with values between 0.6 and 1 can be obtained (*Figure 3A*). The slopes of experimental data range between 0.84 and 0.99, we therefore conclude that linear noise is a relatively good approximation to perform stochastic modeling of microbial communities.

## The strength of the noise determines the width of the distribution of ratios $x(t + \delta t)/x(t)$

Next, we examine the distribution of the ratios of abundances at successive time points (see *Box 1E*). As expected, for significant noise, this distribution can be approximated by a lognormal curve and the width of the distribution becomes larger for increasing noise strength (*Figure 3B*). In order to have widths that are of the same order of magnitude as the ones of the experimental data, the noise must be sufficiently strong. Another way of increasing the width is through interactions, this effect is only moderate. These results are presented in *Supplementary file 1*: Supporting results.

## Stochastic logistic models capture the properties of experimental time series

By using all previous results and imposing the steady-state of experimental data, we find that it is possible to generate time series with identical characteristics to the ones seen in the experimental time series (*Figure 4*). Furthermore, these time series can be generated without introducing any interaction between the different species, but their neutrality measures can still be in the niche regime (*Figure 4F*). Out of 100 simulations, 62 had a p-value smaller than 0.05 for the neutral

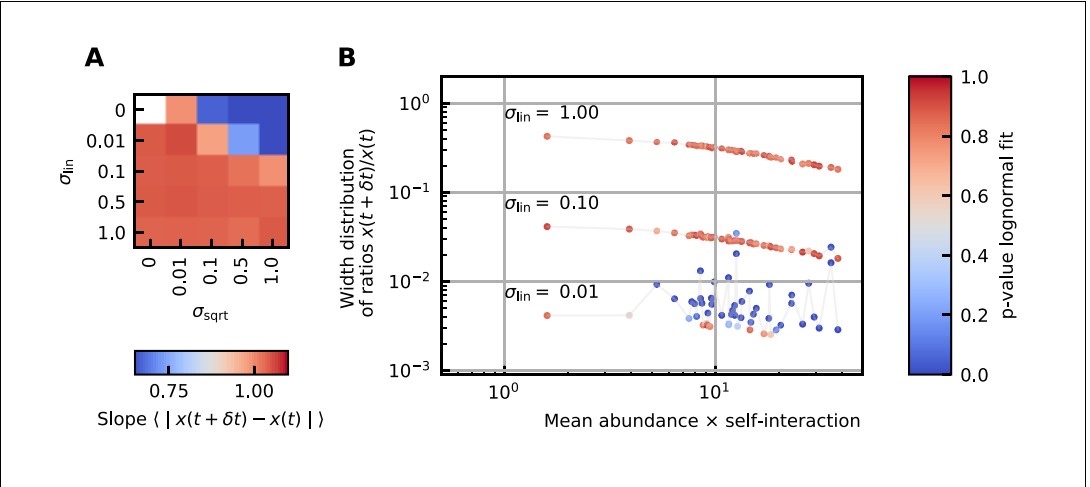

**Figure 3.** Differences between time points as a function of the noise. (**A**) Correlation between the mean absolute differences between abundances at successive time points and the mean abundance for different strengths of the linear noise ($\sigma_{lin}$) and multiplicative noise that scales with the square root of the abundances ($\sigma_{sqrt}$). More specifically, the parameter represents the slope of the logarithm of the mean absolute difference between abundances at successive time points as a function of the logarithm of the mean abundance. Examples of such slopes are given by *Figure 1D*. Here, the slope ranges from 0.66 for noise that scales with the square root to one for linear noise. (**B**) The width of the distribution of the ratios of abundances at successive time points increases for increasing strength of the noise. For sufficiently strong noise the distribution is well fitted by a lognormal function (high p-values for the Kolmogorov-Smirnov test).

covariance test which means they are in the niche regime. The colors of the noise fix the self-interaction values (*Figure 4C*), next the rank abundance distribution is imposed by calculating the growth vector $\hat{g}$ (*Figure 4B*). The slope of the curve of the mean absolute difference between abundances at successive time points as a function of the mean abundance is one by using linear multiplicative noise (*Figure 4D*) and the width of the fluctuations is tuned by choosing a large noise size $\sigma$ (*Figure 4E*). In most experimental time series, only the fractional abundances of species can be measured per time point and not the absolute ones. Because the total abundance of all species remains nearly constant in time series generated by a stochastic logistic equation, our results still hold for time series with fractional abundances (see Supporting results). Similar results can be obtained for models with interactions (see Supporting results), but we want to stress that interactions are not needed to reproduce the properties of experimental time series.

## Discussion

Recent research has focused on different aspects of experimental time series of microbial dynamics, in particular the rank abundance distribution, the noise color, the stability, and neutrality. Within the framework of stochastic generalized Lotka-Volterra models, we studied the influence of growth rates, interactions between species, and the different sources of stochasticity on the observed characteristics of the noise and on neutrality. Our observations are:

- Even when we consider the case without interactions between species, the result of the neutrality test on the time series is often niche. We should, therefore, be careful in the interpretation of the results of neutrality tests.
- For a given sampling step $\delta t$, the noise color depends on the product of the self-interaction and the mean abundance, which for noninteracting species reduces to a dependence on the growth rate. Assuming the model can be used for microbial communities, the self-interaction coefficients can be estimated given the mean abundance, noise color, and sampling rate. Low sampling rates result in larger errors (*Figure 2B*). For sparsely sampled experimental data, the

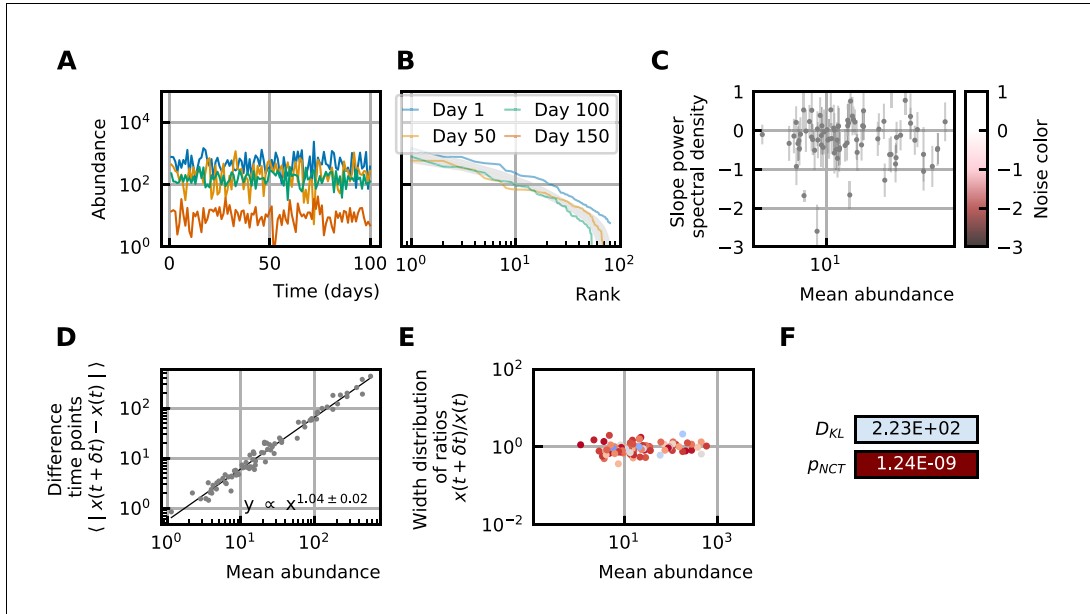

**Figure 4.** A stochastic logistic model is able to reproduce the different characteristics of the noise. (**A**) Time series. (**B**) A rank abundance that remains stable over time. (**C**) Results of the neutrality test in the niche regime. (**D**) Noise color in the white-pink region with no dependence on the mean abundance. (**E**) The slope of the mean absolute difference between abundances at successive time points is around 1. (**F**) The width of the distribution of the ratios of abundances at successive time points is in the order of 1 and independent of the mean abundance.

standard deviation of the self-interaction inferred using the noise color will be larger. For the experimental time series (plankton, gut, and human microbiome) the self-interaction strengths range over several orders of magnitude. The convention of equalling all self-interactions to $-1$ used in several studies (**Fisher and Mehta, 2014**; **Gibson et al., 2016**), cannot be adopted for stochastic models of communities with a heavy-tailed abundance distribution.

- The exponent of the mean absolute differences between abundances at successive time points with respect to the mean abundances is slightly smaller than one for experimental time series. Linear multiplicative noise results in a value of one, square root noise results in lower values (0.6). A mix of linear and square root noise can result in slopes with intermediate values.
- A large multiplicative linear noise is in agreement with both the distribution of the ratios of abundances at successive time points and the relation between the differences between abundances at successive time points and mean values.

To conclude, characteristics of experimental time series, from plankton to gut microbiota, can be reproduced by stochastic logistic models with a dominant linear noise. We expect, however, that for higher sampling rates, modeling the interactions between microbes would be necessary to explain the properties of the time series. For gut microbial time series, the system is sampled only once a day and therefore dominated by the noise in the growth terms corresponding to a linear noise.

Predictive models for the dynamics of microbial communities will certainly require a more in-depth description of the system. Nutrients and spatial distribution of microbes should play a role to dictate the evolution of the community, as well as the interaction with the environment. Synthetic microbial communities are currently being developed and will hopefully provide a more comprehensive view on the complexity of microbial communities (**Vrancken et al., 2019**).

## Materials and methods

### Modeling generalized Lotka-Volterra equations

In a microbial community different species interact because they compete for the same resources. Moreover, they produce byproducts that can affect the growth of other species. Depending on the nature of the byproducts, harmful, beneficial, or even essential, the interaction strength will be either negative or positive. To describe the dynamics of interacting species, one can use the generalized Lotka-Volterra equations:

$$\dot{x}_i = \lambda_i + g_i x_i + \sum_j \omega_{ij} x_i x_j, \tag{1}$$

where $x_i$, $\lambda_i$ and $g_i$ are the abundance, the immigration rate, and the growth rate of species $i$ respectively, and $\omega_{ij}$ is the interaction coefficient that represents the effect of species $j$ on species $i$. The diagonal elements of the interaction matrix $\omega_{ii}$, the so-called self-interactions, are negative to ensure stable steady-states. The off-diagonal elements of the interaction matrix $\omega_{ij}$ are drawn from a normal distribution with standard deviation $\alpha$ ($\omega_{ij} \sim \mathcal{N}(0, \alpha^2)$). The gLV equations only consider pairwise effects and no saturation terms, or other higher-order terms. Due to this drawback, these models sometimes fail to predict microbial dynamics (*Momeni et al., 2017*; *Levine et al., 2017*). However, they are among the most simple models for interacting species and therefore widely studied and used. Noninteracting species can be described by the logistic model, which is a special case of the gLV model obtained by setting all off-diagonal elements of the interaction matrix to zero.

### Implementations of the noise

There exist two principal types of noise: intrinsic and extrinsic noise. *Extrinsic noise* arises due to external sources that can alter the values of the different variables: the immigration rate and growth rate fluctuate in time through colonization of species or a changing flux of nutrients. These processes give rise to additive and linear multiplicative noise respectively. The remaining parameters, inter- and intra-specific interactions can also, change depending on the environment. The formulation of this noise is more subtle (used in *Zhu and Yin, 2009*). *Intrinsic noise* is due to the discrete nature of individual microbial cells. Thermal fluctuations at the molecular level determine the fitness of the individual cells. Therefore, cell growth, cell division, and cell death can be considered as stochastic Poisson processes. For large numbers of microbes, these fluctuations will be averaged out.

We first consider the extrinsic noise. If the time series is calculated by $x_i(t + dt) = x_i(t) + dx_i(t)$, the implementation of the linear multiplicative noise is as follows,

$$dx_i(t) = \lambda_i dt + g_i x_i(t) dt + \sum_j \omega_{ij} x_i(t) x_j(t) dt + x_i(t) \sigma_i dW(t), \tag{2}$$

where $dW$ is an infinitesimal element of a Brownian motion defined by a variance of $dt$ ($dW \sim \sqrt{dt} \mathcal{N}(0, 1)$). Changes in immigration rates of microbial species can be modeled with additive noise,

$$dx_i = \lambda_i dt + g_i x_i dt + \sum_j \omega_{ij} x_i x_j dt + \sigma_{i,\text{const}} dW_{\text{const}}, \tag{3}$$

with $dW_{\text{const}} \sim \sqrt{dt} \mathcal{N}(0, 1)$. Our main motivation is to model the gut microbiome in the colon. Here, we ignore the immigration of species for two reasons. First, the number of microbes in the colon is orders of magnitude larger than the number of microbes in the other parts of the gut (*Marteau et al., 2001*; *Gorbach et al., 1967*)—therefore, the flux of incoming microbes in the colon is small. Second, we only consider systems around steady-state, for which we assume immigration does not play an important role. For perturbed systems, which are far from equilibrium, immigration rates cannot be ignored. Ignoring immigration may be too restrictive for some microbial systems such as the skin microbiome or plankton.

To derive the form of intrinsic noise in generalized Lotka-Volterra equations, we can consider every species abundance making a random walk in one dimension. The average displacement is zero and the variance of displacement is the sum of the rate of growth (jumping to the right) and the rate of death (jumping to the left). For the generalized Lotka-Volterra equations, this results in a noise term

$$\langle n_i(t)n_i(t')\rangle = (\text{growth rate}_i + \text{death rate}_i)x_i = (f(g_i) + h(\omega,\vec{x}))x_i\delta(t-t'), \tag{4}$$

with $\omega$ the interaction matrix and where functions $f$ and $h$ each decouple the growth and death terms. In the generalized Lotka-Volterra model no difference is made between negative interactions as a result of slowing down the growth rate or increasing the death rate, only the resulting net rates are used. This distinction must however be made to implement the intrinsic noise for gLV. In our analysis, we use the simpler logistic models where the resulting variance of the noise is proportional to the square root of the abundance $\sqrt{x}$. One must be careful not to use this noise for values that are smaller than one because this derivation relies on Poisson statistics which is defined for integer numbers.

We implement the intrinsic noise by a term that scales with the square root of the species abundance (**Walczak et al., 2012**; **Fisher and Mehta, 2014**),

$$dx_i(t) = \lambda_i dt + g_i x_i(t)dt + \sum_j \omega_{ij}x_i(t)x_j(t)dt + \%\sqrt{x_i(t)}\sigma_{i,\text{sqrt}}dW_{\text{sqrt}}, \tag{5}$$

with $dW_{\text{sqrt}}$ again an infinitesimal element of a Brownian motion defined by a variance of $dt$ ($dW_{\text{sqrt}} \sim \sqrt{dt}\mathcal{N}(0,1)$). The size of this noise $\sigma_{i,\text{sqrt}}$ is determined by the cell division ($g^+$) and death rates ($g^-$) separately, which are in our model combined to one growth vector ($g = g^+ - g^-$, $\sigma_{i,\text{sqrt}} = \sqrt{g^+ + g^-}$), for large division and death rates the intrinsic noise will be larger.

To sum up, we focus on linear multiplicative noise because: (a) extrinsic noise is dominant as microbial communities contain a very large number of individuals and (b) we ignore the immigration of individuals in our analysis.

We verified that our analysis is robust with respect to the multiple possibilities for the discretization of these models. We also compare our population-level approach with individual-based modeling approaches. Details can be found in the **Supplementary file 1**: Supporting results.

## Neutrality measures

There is no consensus on the definition of neutrality. In general, ecosystems are considered neutral if the dominating cause of fluctuations is random birth and death processes and not fitness advantages of species.

Different neutrality measures focus on different aspects of neutrality. The Kullback-Leibler divergence verifies whether all species are equal (equal abundances and equal covariances). The neutrality covariance test studies the grouping invariance of species in time series.

Given two distributions $P$ and $Q$, the *Kullback-Leibler divergence* is defined as

$$D_{KL}(P|Q) = E_P\left[\ln\frac{P}{Q}\right] \tag{6}$$

where $E_P$ is the expectation value using the probabilities of distribution $P$. The density function of a multivariate Gaussian distribution is

$$P(x) = \frac{1}{(2\pi)^{N/2}\sqrt{\det K}}\exp\left(-\frac{1}{2}(x-\mu)^T K^{-1}(x-\mu)\right) \tag{7}$$

where $\mu$ and $K$ are the mean and covariance matrix of the distribution respectively. The Kullback-Leibler divergence for two multivariate Gaussian distributions in $\mathbb{R}^n$ is (**Duchi, 2007**)

$$D_{KL}(P|Q) = \frac{1}{2}\left(\ln\frac{\det K_Q}{\det K_P} - n + \text{Tr}\left(K_Q^{-1}K_P\right) + (\mu_Q - \mu_P)^T K_Q^{-1}(\mu_Q - \mu_P)\right). \tag{8}$$

For every time series, we can calculate the mean $\mu$ and covariance matrix $K$, and define values $\mu_N$ and $K_N$ for a corresponding neutral time series in which all species are equal (**Fisher and Mehta,**

*2014*). The distance to neutrality $D_{KL}(P|P_N)$ can thus be calculated by computing the probability distribution of the original time series $P$ and the associated neutral distribution $P_N$ with mean values $\mu_N = S^{-1}\sum_{i=1}^{S}\mu_i$ and $K_{P,ii} = S^{-1}\sum_{i=1}^{S}K_{ii}$ and $K_{P,ij} = S^{-1}(S-1)^{-1}\sum_{i=1}^{S}\sum_{j=1,i\neq j}^{S}K_{ij}$ with $S$ the number of species.

The *neutral covariance test* was designed by *Washburne et al., 2016*. We used a python translation of the code developed by this author.

## Noise color

The color of the noise in a time series is determined by the slope of the power spectral density in a log-log scale. This slope can be determined by a linear fit through the spectrum. A different technique to estimate this slope has been put forward by *Faust et al., 2018*. There, it is argued that the power spectral density does not have a constant slope and that, therefore, a nonlinear curve must be fitted. They choose a spline fit and consider the minimal value of its derivative to be the value of the noise color. Because the minimal value of the slope of the fit is taken, the noise color tends to be darker when using this technique. For our time series, however, we see that the spline fit only deviates from the linear fit for low frequencies (*Figure 5*). We ignore the low frequencies for fitting because of the windowing effect. Therefore, we opt for a linear fit after omitting the values for low frequencies (one order of magnitude of the lowest frequencies).

The correspondence between the colors and slopes is here:

| Slope | Color |
| --- | --- |
| 0 | white |
| -1 | pink |
| -2 | brown |
| -3 | black |

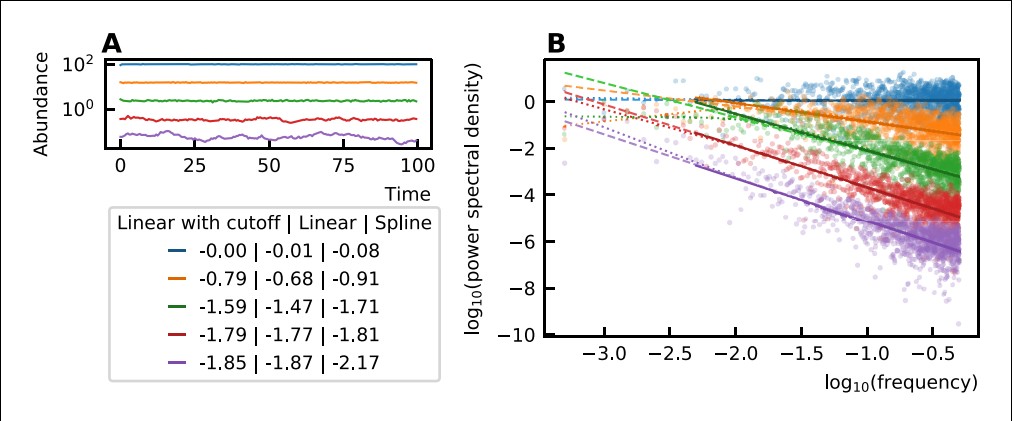

**Figure 5.** The noise color of time series (**A**) is determined by the slope of the power spectral density (**B**). This slope can be measured through a linear fit of all values (dashed), a linear fit through the higher frequency range (solid line) or by performing a spline fit (dotted). A linear fit through all frequencies can be influenced by the windowing effect for low frequencies and the spline fit can make the slope steeper at the low frequencies and result in a darker noise as can be seen for the purple curves. The values of the noise color determined by the different techniques are given in the legend. Therefore, in our work, we opt for the linear fit with a cutoff for low frequencies.

## Acknowledgements

We thank Karoline Faust for interesting discussions, Pankaj Mehta for clarifications about his work on the niche-neutral transition and Stefan Vet and Lendert Gelens for careful reading of the manuscript.

## Additional information

### Funding

| Funder | Grant reference number | Author |
|---|---|---|
| Vrije Universiteit Brussel | SRP31 | Lana Descheemaeker |

The funders had no role in study design, data collection and interpretation, or the decision to submit the work for publication.

### Author contributions

Lana Descheemaeker, Conceptualization, Formal analysis, Visualization, Methodology, Writing - original draft, Writing - review and editing; Sophie de Buyl, Conceptualization, Supervision, Funding acquisition, Methodology, Writing - original draft

### Author ORCIDs

Lana Descheemaeker (iD) https://orcid.org/0000-0002-8732-7051
Sophie de Buyl (iD) https://orcid.org/0000-0002-3314-9616

### Decision letter and Author response

Decision letter https://doi.org/10.7554/eLife.55650.sa1
Author response https://doi.org/10.7554/eLife.55650.sa2

## Additional files

### Supplementary files

• Supplementary file 1. Supplemental information. **Analysis of all experimental data**. Rank abundance, distribution of the differences between abundance at successive time points, neutrality tests and noise color. **Supporting results**. Code All python codes to perform time series simulations, analysis and make all different figures of the main paper and supplement are available at https://github.com/lanadescheemaeker/logistic_models (*Descheemaeker and de Buyl, 2020*; copy archived at https://github.com/elifesciences-publications/logistic_models).

• Transparent reporting form

### Data availability

All data used in this study is available at https://github.com/lanadescheemaeker/logistic_models (copy archived at https://github.com/elifesciences-publications/logistic_models).

The following previously published datasets were used:

| Author(s) | Year | Dataset title | Dataset URL | Database and Identifier |
|---|---|---|---|---|
| David LA, Materna AC, Friedman J, Campos-Baptista MI, Blackburn MC, Perrotta A, Erdman SE, Alm EJ | 2014 | Host lifestyle affects human microbiota on daily timescales | https://www.ebi.ac.uk/ena/data/view/PRJEB6518 | EBI/ENA database, PRJEB6518 |
| Caporaso JG, Lau- | 2011 | Moving Pictures of the Human | https://www.mg-rast.org/ | MG-RAST, mgp93 |

| ber CL, Costello EK, Berg-Lyons D, Gonzalez A, Stombaugh J, Knights D, Gajer P, Ravel J, Fierer N, Gordon JI, Knight R | Microbiome | mgmain.html?mgpage= project&project=mgp93 |
|---|---|---|

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
