## [Decision Letter]

**Acceptance summary:**

This study shows that a very simple non-interacting model, with an appropriate choice of noise, can explain various summary statistics of microbial community time series. This serves as an important null hypothesis, that can be used to rebut over-interpretations of such time series data when they are used to infer underlying interactions.

**Decision letter after peer review:**

Thank you for submitting your article "Stochastic logistic models reproduce experimental time series of microbial communities" for consideration by *eLife*. Your article has been reviewed by Aleksandra Walczak as the Senior Editor, a Reviewing Editor, and two reviewers. The reviewers have opted to remain anonymous.

The reviewers have discussed the reviews with one another and the Reviewing Editor has drafted this decision to help you prepare a revised submission.

Summary:

Both reviewers found points of interest in your manuscript, and appreciated the usefulness of a simple model that is consistent with the data. However, both have also raised some important issues which need to be addressed, which range from clarity of the presentation, to questions about the methods, and robustness of the results. Please address each of the issues raised below by the reviewers in your revised manuscript.

In addition, it is clear from the reviews and our reading that you have not ruled out other more complex models (e.g. models with interactions). Therefore, we would like you to remove or rewrite statements that make overly strong claims about the real systems, and be more clear which claims are about your model and which about the real system. For instance, the last sentence of the Abstract – "This suggests that fluctuations in the sparsely sampled experimental time series are caused by extrinsic sources." – is too strong a claim to make about the real system. Another example from the Abstract is "We show that the noise term should be large and that it is a linear function of the species abundance, while the strength of the self-interactions should vary.…", which should be modified to make it clearer that these are requirements for the model to fit the data.

Reviewer #1:

Deschmeemaeker et al., use a stochastic generalized Lotka-Volterra model to demonstrate that the properties of human microbes can be explained using a logistic modeling approach. Moreover, their results suggest that the niche state of community dynamics can be explained by the intraspecies fluctuations!, this outcome I find particularly interesting. The manuscript highlights a novel approach to studying complex microbial communities and the findings are generally interesting. However, I would suggest some changes. Mostly, these concerns can be considered minor, but answering them is crucial for more transparency and clarity on the work presented in this manuscript.

1) Palm and tongue, as well as gut microbiomes, change over time. I would have imagined that these changed cause changes in the rank abundance profile of a community over time. The data, therefore, is interesting (because it is derived from human-associated microbial communities), though not surprising.

2) Readers will benefit if the authors can include information from the methods sections. Such as definitions of noise, noise color, abundance and time series can be included in the main text. This change will help the message from this manuscript to be more accessible to a broader audience.

3) Since the frequency of sampling has an effect on noise color. Can authors elaborate a bit more ( subsection “The self-organized instability model can be reproduced by the stochastic generalized Lotka-Volterra model”) on how the limitation of having experimental time series from only a handful of time points might influence their inference?

4) Statistics: Figure 1(A-E), Figure 4(A,B,D,E,F) error not mentioned.

5) Figure 1A, legends for this figure is not clear to me. These details are part of the main text however it is important that legends of the figure are described in greater detail. For example, what is the meaning of different colors in the time series? Are these randomly selected species? or species with specific ranks are selected for this analysis?

Reviewer #2:

This study aims to find the simplest model which can explain various summary statistics of microbial community time series. The key result is that various statistics of interest can be obtained using a non-interacting logistic model with appropriate choice of noise. This serves as a form of null hypothesis, and could be used to rebut over-interpretations of such time series.

Overall, I am sympathetic to the goals of the study. I have the following suggestions:

Specific comments:

1) For non-specialists it would be good to start with a summary of the neutral vs niche regime, as this point is rather central to the discussion. (If the time series seem to be operating in the niche regime, then it is not surprising per se that a non-interacting model can capture some summary statistics of the time series.)

2) This comes across mainly a study of a model, rather than a study of datasets which produces specific insights into microbial communities of different types. This is partly the point of the paper, that the real time series are somehow not sufficient to reveal the true interactions. Of course, the interactions in any real system are neither zero, nor equal, nor randomly sampled, but can be discerned with sufficiently sensitive experiments. So one expects that, after full study of this type, the non-interacting model proposed by the authors will not be supported. The authors do note that interactions cannot be ruled out in their approach. But could some text be included to say what the reader should take away from the study: is it meant to be a null model, or predictive?

3) The results are obtained for a particular choice of growth rates to match given abundances. This particular choice is indeed shown to replicate the observed time series summary statistics. However, it does mean that the other observations made by the authors (specific relations of noise / abundances etc.) could depend on this growth rate choice. How robust are the various observations to changing the particular mechanism of setting abundances?

4) I had difficulty distinguishing between terms "dt", "*δ*t", "*δ*", "delay", and "sampling time" which the authors seem to use interchangeably through the text and supplement. For example, in Figure 2B should it be "Sampling dt" or "Sampling *δ*t"? In Figure S4 they use the term "*δ* = 0" which I think should be "*δ*t = 0"? The term "time delay" is not defined elsewhere. More broadly, in a correct implementation of a stochastic simulation, the result should be asymptotically independent of integration timestep. The authors provide detailed comparison of different discretizations, this is appreciated (Figure S5). However, in the supplement I could not see how the "dt" in the SDE is related to the integration time for numerical solution of the Langevin equation "*δ*t", unless they were the same term. I could not at all follow Figure 2B. Is the "sampling time" different from the numerical integration timestep? Sampling is correctly done by first integrating with a small timestep and showing results independent of timestep, and then sampling at the desired sampling intervals. If the sampling is related to the bandwidth limitation, that should be analyzed separately from the numerical integration.

---

## [Author Response]

Summary:Both reviewers found points of interest in your manuscript, and appreciated the usefulness of a simple model that is consistent with the data. However, both have also raised some important issues which need to be addressed, which range from clarity of the presentation, to questions about the methods, and robustness of the results. Please address each of the issues raised below by the reviewers in your revised manuscript.In addition, it is clear from the reviews and our reading that you have not ruled out other more complex models (e.g. models with interactions). Therefore, we would like you to remove or rewrite statements that make overly strong claims about the real systems, and be more clear which claims are about your model and which about the real system. For instance, the last sentence of the Abstract – "This suggests that fluctuations in the sparsely sampled experimental time series are caused by extrinsic sources." – is too strong a claim to make about the real system. Another example from the Abstract is "We show that the noise term should be large and that it is a linear function of the species abundance, while the strength of the self-interactions should vary.…", which should be modified to make it clearer that these are requirements for the model to fit the data.

We changed the overly strong claims of the Abstract.

Reviewer #1:Deschmeemaeker et al., use a stochastic generalized Lotka-Volterra model to demonstrate that the properties of human microbes can be explained using a logistic modeling approach. Moreover, their results suggest that the niche state of community dynamics can be explained by the intraspecies fluctuations!, this outcome I find particularly interesting. The manuscript highlights a novel approach to studying complex microbial communities and the findings are generally interesting. However, I would suggest some changes. Mostly, these concerns can be considered minor, but answering them is crucial for more transparency and clarity on the work presented in this manuscript.1) Palm and tongue, as well as gut microbiomes, change over time. I would have imagined that these changed cause changes in the rank abundance profile of a community over time. The data, therefore, is interesting (because it is derived from human-associated microbial communities), though not surprising.

Indeed, although the species abundances fluctuate over time, the rank abundance profile remains stable over time, as can be seen in Figure 1B. A comment to acknowledge this fact was added to the text (Results section).

2) Readers will benefit if the authors can include information from the methods sections. Such as definitions of noise, noise color, abundance and time series can be included in the main text. This change will help the message from this manuscript to be more accessible to a broader audience.

We added sentences introducing the concepts ’abundance’ and noise in the Introduction. The technical aspects of noise can be found in subsection “Implementations of the noise”. We also created a feature box with the definitions of all the concepts, such that the reader can find this information easily. The features are labeled with the letters of Figure 1 and Figure 4, of which we changed the order to remain consistent.

3) Since the frequency of sampling has an effect on noise color. Can authors elaborate a bit more (subsection “The self-organized instability model can be reproduced by the stochastic generalized Lotka-Volterra model”) on how the limitation of having experimental time series from only a handful of time points might influence their inference?

Low sampling rates result in larger errors (Figure 2B). For sparsely sampled experimental data, the standard deviation of the self-interaction inferred using the noise color will be larger (Discussion section).

4) Statistics: Figure 1(A-E), Figure 4(A,B,D,E,F) error not mentioned.

We did not represent experimental error bars on Figure 1(A-B) as this data is not available. We expect the errors due to measurements or sampling to be smaller than the fluctuations due to biological processes. This hypothesis is supported by the results of [Silverman et al., 2018] for microbial communities of an artificial gut. Here, the biological variation becomes five to six times more important than the technical variation for the sampling interval of a day (Figure 3B of [Silverman et al., 2018]). Also, [Grilli, 2019] shows the time correlation of experimental time series which is non-zero while if technical variation is the main source of variation, no correlation should be noticeable. We added a paragraph in the manuscript about these errorbars (Results section). For Figure 1C, we added the error of the slope of the power spectral density (noise color). For Figure 1D, we added the error on the exponents. For Figure 1E, the p-value corresponds to the Kolmogorov-Smirnov test. Concerning Figure 4, similar errors have been added. We did not include errors for Figure 4(A-B) as those figures correspond to one simulation.

5) Figure 1A, legends for this figure is not clear to me. These details are part of the main text however it is important that legends of the figure are described in greater detail. For example, what is the meaning of different colors in the time series? Are these randomly selected species? or species with specific ranks are selected for this analysis?

In the first submission, we selected species with evenly spread ranks. In the figure of this submission, we used another selection of species with ranks 1, 5, 10, 50 and 100. We added a legend denoting the rank number of all species.

Reviewer #2:This study aims to find the simplest model which can explain various summary statistics of microbial community time series. The key result is that various statistics of interest can be obtained using a non-interacting logistic model with appropriate choice of noise. This serves as a form of null hypothesis, and could be used to rebut over-interpretations of such time series.Overall, I am sympathetic to the goals of the study. I have the following suggestions:Specific comments:1) For non-specialists it would be good to start with a summary of the neutral vs niche regime, as this point is rather central to the discussion. (If the time series seem to be operating in the niche regime, then it is not surprising per se that a non-interacting model can capture some summary statistics of the time series.)

We added extra sentences about the concept of niche and neutrality (Introduction). A negative neutrality test is sometimes used to state that there must be interactions between species which are subsequently inferred, as for instance in [Faust et al., 2018]. We say that a noninteracting model can be classified in the niche regime by a neutrality test.

2) This comes across mainly a study of a model, rather than a study of datasets which produces specific insights into microbial communities of different types. This is partly the point of the paper, that the real time series are somehow not sufficient to reveal the true interactions. Of course, the interactions in any real system are neither zero, nor equal, nor randomly sampled, but can be discerned with sufficiently sensitive experiments. So one expects that, after full study of this type, the non-interacting model proposed by the authors will not be supported. The authors do note that interactions cannot be ruled out in their approach. But could some text be included to say what the reader should take away from the study: is it meant to be a null model, or predictive?

The logistic model with linear noise can be considered as a null model (Introduction). We do not claim that there are no interactions between species. We show that logistic models are able to reproduce most of the characteristics of experimental time series, what implies that no interactions are needed in the model to obtain the statistical properties.

3) The results are obtained for a particular choice of growth rates to match given abundances. This particular choice is indeed shown to replicate the observed time series summary statistics. However, it does mean that the other observations made by the authors (specific relations of noise / abundances etc.) could depend on this growth rate choice. How robust are the various observations to changing the particular mechanism of setting abundances?

For logistic models, the growth rate is equal to the product of the self-interaction and mean abundance. The noise color and the width of the distribution of ratios *x*(*t* + *δt*)*/x*(*t*) depend on this product. In order to obtain given characteristics—a predefined noise color and width of the distribution of ratios *x*(*t*+*δt*)*/x*(*t*)—the choice of the growth rate will therefore dictate the choice of the remaining free parameters, the sampling time step *δt* and the noise strength *σ*. This clarification was added to the main paper (Results section).

4) I had difficulty distinguishing between terms "dt", "δt", "δ", "delay", and "sampling time" which the authors seem to use interchangeably through the text and supplement. For example, in Figure 2B should it be "Sampling dt" or "Sampling δt"? In Figure S4 they use the term "δ = 0" which I think should be "δt = 0"? The term "time delay" is not defined elsewhere. More broadly, in a correct implementation of a stochastic simulation, the result should be asymptotically independent of integration timestep. The authors provide detailed comparison of different discretizations, this is appreciated (Figure S5). However, in the supplement I could not see how the "dt" in the SDE is related to the integration time for numerical solution of the Langevin equation "δt", unless they were the same term. I could not at all follow Figure 2B. Is the "sampling time" different from the numerical integration timestep? Sampling is correctly done by first integrating with a small timestep and showing results independent of timestep, and then sampling at the desired sampling intervals. If the sampling is related to the bandwidth limitation, that should be analyzed separately from the numerical integration.

We are sorry for the confusion. We use “dt” for the integration time step and “*δ*t” for the sampling step, i.e. the time step in the time series which is a multiple of the integration time step. Figure 2B was therefore wrongly labeled and we changed this in the new version of the manuscript. We integrated with small time steps *dt*, which we previously called the fundamental time steps but are now labeled as integration time steps for clarity. We compared our results for different values of this integration time step *dt* (Figure 5E of the supplement) and concluded that our chosen time step *dt* was small enough such that there was no dependence on this step size. The terminology of “delay” and “*δ*” was introduced in the context of the autocorrelation function which is defined as a function of time delay.